# Burdens of non-communicable disease attributable to metabolic risk factors in Australia, 1990–2019: joinpoint regression analysis of the Global Burden of Disease Study

Kelemu Tilahun Kibret  , Kathryn Backholer, Anna Peeters, Fisaha Tesfay, Melanie Nichols

Global Centre for Preventive Health and NutritionI, Institute for Health Transformation, School of Health and Social Development, Faculty of Health, Deakin University, Geelong, Victoria, Australia

**Correspondence to**
Dr Kelemu Tilahun Kibret;
kelemu.kibret@deakin.edu.au

## ABSTRACT

**Background** Long-term and comparative assessments of trends in non-communicable disease (NCD) burden attributable to metabolic risk are sparse. This study aimed to assess burdens and trends of NCD attributable to metabolic risk factors in Australia, 1990–2019.

**Design** Population-based observational study.

**Settings and data source** Data were extracted from the Global Burden of Disease Study 2019 for Australia and trends in NCD burden attributable metabolic risks were estimated using the joinpoint regression model.

**Main outcome measures** NCD deaths and disability-adjusted life-years (DALYs) attributed to metabolic risk factors, 1990–2019.

**Results** Results indicate a 1.1% yearly increase in exposure to combined metabolic risk factors from 1990 to 2019. Between 1990 and 2019, the estimated absolute number of deaths from NCDs attributed to combined metabolic risks increased by 17.0%. However, metabolic risk-related NCD burdens in Australia decreased between 1990 and 2019. In 2019, 34.0% of NCD deaths and 20.0% of NCD DALYs were attributed to metabolic risk factors, compared with 42.9% and 24.4%, respectively, in 1990. In 2019, cardiovascular diseases (CVDs), neoplasms and chronic kidney diseases were the most common NCD deaths attributed to metabolic risks. High body mass index accounted for the highest proportion of diabetes deaths (47.0%) and DALYs (58.1%) as well as chronic kidney disease deaths (35.4%) and DALYs (39.7%). Similarly, high systolic blood pressure contributed to a high proportion of chronic kidney disease deaths (60.9%) and DALYs (53.2%), and CVDs deaths (44.0%) and DALYs (46.0%).

**Conclusion** While the contribution of metabolic risk factors to the burden of NCDs has declined from 1990 to 2019, their role in NCD death and disability remains a challenge as the prevalence of these risk factors has increased. Prevention strategies should focus on metabolic risks particularly high body mass index and high systolic blood pressure to substantially reduce NCD burdens.

## STRENGTHS AND LIMITATIONS OF THIS STUDY

⇒ Application of longitudinal statistical method (joinpoint regression analysis) which helps to model the long-term trend of non-communicable disease burden attributed to metabolic risks.

⇒ The use of data from the Global Burden of Disease (GBD) dataset, which is one of the most comprehensive and reliable data sources.

⇒ The likelihood that the metabolic risk-related disease burden may vary between rural and urban areas, different regions or states and sociocultural conditions, which could not be characterised in this study.

⇒ In GBD, the use of similar relative risks in certain age and gender groups in all countries may overestimate or underestimate the exact picture of disease burden, because metabolic risk factors may not influence disease outcomes in the same way in different population groups.

## INTRODUCTION

Currently, non-communicable diseases (NCDs) are the leading cause of mortality and morbidity worldwide, accounting for 71.0% of all deaths.[1] In particular, cardiovascular diseases (CVDs) (eg, ischaemic heart disease and stroke) are leading causes of death in both high-income and low-income and middle-income countries.[2][3] NCDs are multifactorial diseases caused by complex interactions of both behavioural and biological (metabolic) risk factors.[4] Metabolic risk factors, including high systolic blood pressure (SBP), high body mass index (BMI) and high fasting plasma glucose (FPG), are among the top five leading risk factors (with smoking and unhealthy diet) for NCDs in the world.[4] Globally, high SBP (SBP≥130 mm Hg), high FPG (≥7 mmol/L) and high BMI (BMI≥25 kg/m$^2$) were, respectively, responsible for 9.3%, 6.8% and 6.3% of disability-adjusted life-years (DALYs) in 2019.[4] The largest proportion of disease burden in Australia in 2018 was attributed to metabolic risk factors—high BMI

(8.4%), high blood pressure (5.1%) and high blood plasma glucose (6.3%).[5]

Globally, the largest increases in the prevalence of modifiable risk factors from 2010 to 2019 were for metabolic risk factors such as high FPG and high BMI.[4] According to the Australian National Health Survey 2017–2018, the prevalence of high BMI (BMI≥25 kg/m$^2$) was 67.0%, high blood pressure (SBP≥130 mm Hg or diastolic blood pressure ≥90 mm Hg was 22.8%, high FPG was 5% in adults aged over 18 years.[6] Further, 21% of individuals aged 65 years and over had high cholesterol level (≥240 mg/dL).[6]

NCDs have been recognised as a global priority in the United Nations' Sustainable Development Goals (SDG), which set a target of reducing premature death due to NCDs by one-third by 2030.[7] Similarly, the WHO has set a global target of a 25% reduction in premature death from CVDs, cancer, type 2 diabetes or chronic respiratory diseases (CRDs) by 2025, which can be achieved by reducing the burden of metabolic and behavioural risk factors using appropriate prevention approaches.[8] A modelling study on the effect of risk factors on NCDs showed that premature mortality from the four main NCDs may be decreased worldwide by 22% in men and 19% in women if the SDG risk factor targets could be accomplished.[9] A modelling study in Australia shows that a 25% relative reduction in premature deaths due to four NCDs in 2025, compared with 2010 levels, could be achieved if the age-standardised mortality rates continue to decrease by an average of 1.64% annually.[10]

A clear understanding of the trends and impacts of metabolic risk factors for NCDs is necessary to prioritise prevention and control strategies to achieve NCD reduction goals. The evidence needs to be updated and tracked using the most recent available data,[8 11] and consistent, long-term and comparable findings at the national level are necessary. Assessing which risk factors notably increased or decreased over time and their impact on major NCDs is essential to prioritise and evaluate public health strategies and programmes at the regional or national level. Australia is a high-income country that has seen an overall decline in CVD mortality rates for some years now,[12] but continues to experience a high prevalence of metabolic risk factors[5]. Australia's situation is not unique—many high-income countries also face similar challenges.[13] For example, the USA, and many European countries also have high levels of obesity[14] and type 2 diabetes.[15] Although some studies have examined the burden of diseases due to high BMI, blood glucose or cholesterol level in Australia covering short periods,[5 16 17] long-term and comparative assessment of trends in NCD burden attributable to metabolic risk is still sparse. It is unclear which specific NCD burden attributed to metabolic risks has increased or decreased over time and for which metabolic risk factor. Thus, this study aims to investigate trends in the burden of NCDs attributable to metabolic risk factors in Australia between 1990 and 2019 using the latest Global Burden of Diseases (GBD) study data. The objectives are to assess: (1) changes in exposure

levels of metabolic risk factors in Australia between 1990 and 2019; (2) the burden of NCDs (deaths and DALYs) attributable to metabolic risk factors in 2019; (3) temporal trends in the burden of NCDs attributable to metabolic risk from 1990 to 2019. This study provides context-specific insights into the burden of NCDs attributable to key metabolic risk factors in Australia. The study demonstrates the changing trends in this burden of NCDs attributable to metabolic risk factors over time. Our findings can inform the development of targeted prevention and management strategies for specific population groups, defined by high BMI and high blood pressure.

## METHODS

### Patient and public involvement
Patients or members of the public were not involved in this study.

### Study design and setting
This study is a population-based observational study with an in-depth analysis of the GBD study 2019 data of Australia using a joinpoint regression analysis.

### Data sources
The GBD 2019 study data were used for this study.[4 18] The GBD study is an ongoing global collaborative epidemiological study that comprehensively assesses the burden of diseases, injuries and risk factors for 204 countries and territories by sex and age group, and currently includes data from 1990 to 2019. The GBD provides estimates of deaths and DALYs due to 369 diseases and injuries, and 87 risk factors.[4 18] The GBD study results can be used by global, regional, national, and local policy and decision-makers to gain insights into the burden and trends of population health challenges and optimise responses.

The GBD study incorporates data from many various data sources, including censuses, household surveys, vital statistics, disease registries, disease notifications, health service use, government and international websites and other sources. All the data sources used in GBD are available on the Global Health Data Exchange (GHDx) website (https://ghdx.healthdata.org/gbd-2019/data-input-sources). The collated data are used to generate standard epidemiological estimates, including prevalence, mortality and DALYs, as well as the attributable burdens (mortality and DALYs) for specific risk factors using a comparative risk assessment framework. Through this framework, GBD investigators: (1) identify evidence-based risk–outcome pairs; (2) determine the exposure level associated with minimum risk, termed the theoretical minimum risk exposure level (TMREL) and (3) estimate relative risks, population attributable fractions (PAFs) and attributable burdens (deaths and DALYs) for each risk factor–outcome pair. Detailed descriptions and methods used in GBD 2019 have been published elsewhere,[4 18] and briefly described here.

## Description of variables

### Summary exposure value

The GBD study estimated the exposure distributions for risk factors as a summary exposure value (SEV).[4] SEV is a relative risk-weighted prevalence of exposure, which measures the extent to which a population is exposed to a risk factor and the severity by which that risk contributes to the disease burden. SEV is reported on a scale of 0–100, where 0 means that no excess risk exists for a population, and 100 means that the population is at its highest risk. We assessed the combined metabolic risks (high BMI, high SBP, high FPG and low-density lipoprotein (LDL) cholesterol) and each individual metabolic risk.

### Theoretical minimum risk exposure level

For each risk factor–outcome pair, the GBD study defines what level of exposure to a risk factor leads to the lowest amount of disease.[4] The TMREL value is the level of risk exposure associated with the lowest risk.[4] Detailed TMREL estimation for each risk factor was described elsewhere.[4] In the GBD, the TMREL for 'high BMI' for adults (ages 20+) was defined as BMI 20–25 kg/m$^2$, and for children based on International Obesity Task Force standards.[19] The TMREL level for FPG is defined as 4.8–5.4 mmol/L; and for LDL cholesterol the TMREL defined as 0.7–1.3 mmol/L. The TMREL level for SBP is defined as 110–115 mm Hg.[4]

### Estimation of effect size, PAFs and attributable burdens

The GBD 2019 evaluated the strength of epidemiological evidence on the causal relationship between each metabolic risk factor and disease outcome pair, using the World Cancer Research Fund evidence appraisal tool.[20] For those risk factor–outcome pairs graded as having convincing or probable evidence, relative risks were estimated using data from meta-analysis of cohort, case–control and/or intervention studies.[4] The outcome-specific burden of disease attributable to risk factors was determined by estimating the PAF.[4] PAF is the proportion of the disease burden that would be reduced in a population if exposure to a risk factor was equivalent to the level of TMREL. The PAFs provide estimates of the proportion of cases that might be prevented if a particular risk factor were to be reduced in the population. PAF incorporates the strength of association between risk factors and the outcome of interest, as well as the observed prevalence of risk factors in the population.[21]

In the GBD study, the metrics used to quantify the burden of disease were deaths and DALYs.[3] DALYs are the sum of years of life lost and years lived with disability.[18] For each risk–outcome pair, the attributable burden was estimated by multiplying the numbers of deaths or DALYs observed in the population by the PAF.[4] In the GBD study, age-standardised rates (ASRs) (per 100 000) were estimated using the GBD world standard population, computed using population estimates for 2010–2035 from the World Population Prospects.[18 22] In GBD study, the age standardised proportion is PAF.[4] For some diseases or outcomes in the GBD study, certain risk factors may have a PAF of 100%. The PAF can range from 0% to 100%, but a PAF of 100% does not necessarily mean that the risk factor or exposure is the sole cause of the health outcome or disease. For instance, age-standardised proportion of diabetes mellitus and chronic kidney disease (CKD) deaths was 100% attributed to metabolic risks (high FPG and kidney dysfunction).[4] But it does not necessarily mean that the FPG and kidney dysfunction is the sole cause of diabetes and CKD, respectively.

## Data processing and statistical analysis

This study focused on the burden of NCDs attributable to individual metabolic risk factors (high BMI, high SBP, high FPG and high LDL cholesterol) and combined metabolic risks in Australia. We extracted numbers, proportions and ASRs (per 100 000) of metabolic risk-related deaths and DALYs and their 95% uncertainty intervals (UIs) of NCDs in Australia from 1990 to 2019 from the GBD study's GHDx results tool (http://ghdx.healthdata.org/gbd-results-tool). We selected specific some NCDs (neoplasms, CVDs, diabetes mellitus, CKD, neurological disorders, CRDs) that have summary estimates with common metabolic risk factors. We also extracted NCD deaths and DALYs attributable to metabolic risks by sex and age (20 years and over). Similarly, we extracted the age-standardised SEV for Australia for each metabolic risk factor from 1990 to 2019.

Using these extracted data from GBD results tool, the temporal trends in NCD burdens (deaths and DALYs) attributable to each metabolic risk factor over time (1990–2019) were assessed using the joinpoint regression model (separately by sex).[23] This regression model segments the entire period and identifies the time points (years) at which there has been a significant change in trends (joinpoints).[24] Then, the annual percentage change (APC) with its 95% CI of rates between these inflection points is estimated. The average APC (AAPC) with its 95% CI for the entire period (1990–2019) and for the last decade (2010–2019) was also calculated, with respect to the underlying model fitted by the joinpoint analysis. We examined the trends of the most recent decade for which data were available (2010–2019) to more closely examine the recent trends, which are of policy relevance, and because key policies and strategies related to NCDs have been published in the last decade, such as a Global Action Plan for the Prevention and Control of NCDs[8] and National Primary Health Care Strategy of Australia.[25] AAPC is estimated as a weighted average of the APCs using the joinpoint segment lengths as weights.[26] The AAPC is a summary measure of the trend for the entire period. If there are no joinpoints identified, the model produces only one APC, which is equal to the overall AAPC.

We used the Grid Search Modelling Method with the default recommended parameter values—the minimum number of observations at either end of joinpoint is two, and the minimum number of measurements between two joinpoints is two. We specified a minimum of zero and a

maximum of four joinpoints and a Monte Carlo Permutation method was used to test significant changes in trends over the specified period. The permutation test was also used to determine whether APCs and AAPCs differed from zero using p<0.05 as statistically significant.[24] Data extracted from the GBD were processed in an excel sheet, and the trend analysis was carried out using the Joinpoint Regression Program V.4.9.1.0.[23]

## RESULTS

### Trends in risk exposure for metabolic risk factors in Australia from 1990 to 2019

Most age standardised SEV of individual metabolic risk factors showed an increasing trend; the exposure to combined metabolic risk factors increased by 1.1% per year from 1990 to 2019, though the annual rate of increase was smaller in the most recent decade, at 0.8% from 2010 to 2019. Exposure to high BMI and high FPG showed a yearly increase of 1.5% and 2.0%, respectively, but high SBP decreased by 0.5% per year from 1990 to 2019, with a recent decline in the SEV of high FPG from 2017 and a rise in the SEV of high SBP from 2012 onwards (online supplemental table 1 and online supplemental figure 1).

### The burden of NCDS attributable to metabolic risk factors in Australia

In 2019, metabolic risks accounted for approximately 34.0% of all NCD deaths and 20.0% of NCD DALYs in Australia, with similar age-standardised proportions for both sexes, but a higher proportion of NCD DALYs were attributed to metabolic risks in males than in females (online supplemental table 2). The age-standardised death rates for NCDs linked to metabolic risks were 133.9 per 100 000 for males and 94.7 per 100 000 for females, while the metabolic risk-related age-standardised DALY rate was 3095.8 per 100 000 for males and 2198.7 per 100 000 for females (online supplemental table 2). Age-specific rates and proportion of NCD deaths and DALYs related to metabolic risks increased with age in both sexes (online supplemental figure 2A,B).

Of metabolic risk factors, high SBP and high BMI were the largest contributors to NCD deaths and DALYs, with high SBP estimated to contribute 16.4% of NCD deaths and 8.0% of NCD DALYs, while BMI was estimated to contribute to 12.0% of NCD deaths and 10.0% of NCD DALYs (online supplemental table 3), followed by high FPG which was a significant contributor to NCD deaths (11.4%) and DALYs (7.2%), (online supplemental table 3).

### Temporal trends of metabolic risk-related NCD burden from 1990 to 2019 in Australia

Between 1990 and 2019, the estimated absolute number of NCD deaths and DALYs attributed to combined metabolic risks increased by 17.0% and 13.4%, respectively (online supplemental table 2 and online supplemental figure 3). However, the ASRs of NCD deaths and DALYs

attributable to metabolic risks roughly halved between 1990 and 2019, with AAPCs of −2.6% and −2.1%, respectively (table 1, figure 1A,B).

The proportion of NCD deaths and DALYs had a decreasing trend from 1990 to 2019 (figure 1C,D). The proportions and ASRs of NCD deaths and DALYs related to metabolic risks have decreased for both males and females over time (1990–2019), with higher proportions of metabolic risk-related deaths in females but higher ASRs in males (online supplemental figure 4 A,B, online supplemental table 2). The ASRs and proportions of metabolic risk-related DALYs were also higher in men (online supplemental table 2 and online supplemental figure 4C,D).

Age-standardised NCD death rates attributed to high SBP (AAPC=−3.6%) and high LDL cholesterol (AAPC=−3.9%) showed a significant decrease, resulting in a total decline of 65.4% and 68.4%, respectively, from 1990 to 2019. Similarly, age-standardised NCD DALY rates for high SBP and high LDL cholesterol decreased by an average of 3.6% and 4.1% per year, leading to a total reduction of 65.8% and 70.1% (table 1).

### Specific type of NCD deaths and DALYs attributable to metabolic risk factors

In 2019, CVD deaths (34 921) and DALYs (537 319) attributed to metabolic risks were the highest, followed by neoplasms with 5644 deaths and 110 091 DALYs and CKDs with 5207 deaths and 86 550 DALYs. (online supplemental table 4). The age-standardised proportion of diabetes mellitus and CKD deaths was 100.0% attributed to metabolic risks in Australia. Similarly, a high proportion of age-standardised death rates of CVDs (68.0%), neurological disorders (16.6%) and neoplasms (10.3%) were attributed to combined metabolic risks (table 2).

Each metabolic risk factor was responsible for a sizeable proportion of age-standardised death and DALY rates of specific NCDs (table 2 and online supplemental figure 5). About 47.0% deaths and 58.1% DALYs of diabetes mellitus, and about 35.4% deaths and 39.7% DALYs of CKD were attributed to high BMI. Similarly, 18.6% of CVD deaths and 25.5% of CVD DALYs were attributable to high BMI (table 2). Furthermore, about 18.6% of deaths and 15.5% of DALY for CVDs related to high FPG (table 2). High SBP was responsible for 60.9% deaths and 53.2% DALYs of CKD, and 44.0% deaths and 46.0% DALYs of CVD (table 2).

### Trends of specific NCD deaths and DALYs attributable to metabolic risks

Between 1990 and 2019, the metabolic risk-related age-standardised death rates of CVDs, CRDs and diabetes mellitus significantly declined by an average 3.5%, 3.7% and 0.8% per year (online supplemental table 5). In contrast, age-standardised death rates of neoplasms, neurological disorders and CKD attributed to metabolic risks significantly increased by 0.5%, 1.0% and 0.5% per year, respectively (online supplemental table 5).

**Table 1** The age-standardised death rates and disability-adjusted life-years (DALYs) rates of non-communicable diseases attributable to metabolic risks and their temporal trends between 1990 and 2019 by sex

| Risk factors | Sex | Deaths per 100000 (95% UI) | | | DALYs per 100000 (95% UI) | | |
|---|---|---|---|---|---|---|---|
| | | 1990 | 2019 | AAPC (1990–2019) | 1990 | 2019 | AAPC (1990–2019) |
| Metabolic risk factors | Both | 245.3 (221.7, 266.8) | 113.3 (99.5, 126.6) | −2.6 (−2.8, −2.5)* | 4870.9 (4402.1, 5377.5) | 2630.3 (2246.3, 3063.7) | −2.1 (−2.2, −2.0)* |
| | Male | 302.0 (274.7, 327.6) | 133.9 (118.6, 149.9) | −2.8 (−2.9, −2.6)* | 6124.6 (5580.6, 6704.6) | 3095.8 (2692.8, 3578.5) | −2.3 (−2.4, −2.2)* |
| | Female | 199.8 (177.6, 219.0) | 94.7 (81.5, 107.4) | −2.6 (−2.7, −2.4)* | 3764.7 (3346.0, 4212.5) | 2198.7 (1827.9, 2623.3) | −1.8 (−1.9, −1.7)* |
| High systolic blood pressure | Both | 155.6 (131.2, 177.0) | 53.8 (43.4, 64.4) | −3.6 (−3.8, −3.5)* | 2822.6 (2506.5, 3140.8) | 966.5 (824.7, 1111.5) | −3.6 (−3.8, −3.5)* |
| | Male | 67.9 (47.5, 98.8) | 47.9 (33.0, 67.8) | −3.7 (−3.9, −3.5)* | 1292.4 (1005.1, 1698.2) | 1138.0 (878.9, 1451.8) | −3.7 (−3.9, −3.5)* |
| | Female | 39.1 (27.6, 55.8) | 28.3 (18.7, 42.) | −3.6 (−3.8, −3.4)* | 784.2 (606.7, 1006.7) | 725.1 (538.8, 961.3) | −3.5 (−3.7, −3.4)* |
| High? Body mass index | Both | 65.3 (38.2, 95.6) | 41.7 (27.0, 57.5) | −1.5 (−1.7, −1.3)* | 1801.0 (1102.2, 2556.7) | 1387.7 (954.7, 1855.0) | −0.9 (−0.9, −0.8)* |
| | Male | 75.6 (42.8, 113.0) | 45.9 (27.9, 65.7) | −1.7 (−1.9, −1.6)* | 2091.4 (1252.7, 3014.0) | 1519.1 (1008.5, 2063.2) | −1.1 (−1.1, −1.0)* |
| | Female | 55.1 (33.1, 81.4) | 36.9 (24.4, 50.7) | −1.4 (−1.6, −1.1)* | 1511.5 (947.8, 2146.6) | 1255.7 (874.5, 1670.6) | −0.6 (−0.7, −0.6)* |
| High fasting plasma glucose | Both | 50.2 (35.9, 71.0) | 37.2 (25.7, 53.1) | −0.9 (−1.2, −0.7)* | 1001.0 (782.8, 1275.1) | 919.9 (698.2, 1187.71) | −0.2 (−0.4, −0.1)* |
| | Male | 67.9 (47.5, 98.8) | 47.9 (33.0, 67.8) | −1.2 (−1.3, −1.0)* | 1292.4 (1005.1, 1698.2) | 1138.1 (878.9, 1451.9) | −0.4 (−0.5, −0.2)* |
| | Female | 39.1 (27.6, 55.8) | 28.3 (18.7, 42.1) | −1.0 (−1.3, −0.7)* | 784.2 (606.7, 1006.7) | 725.1 (538.8, 961.3) | −0.3 (−0.5, −0.0)* |
| High LDL cholesterol | Both | 94.9 (70.4, 122.1) | 30.0 (21.1, 40.1) | −3.9 (−4.1, −3.7)* | 1758.0 (1443.6, 2100.7) | 525.2 (420.9, 639.0) | −4.1 (−4.4, −3.8)* |
| | Male | 124.9 (95.7, 157.5) | 38.3 (27.9, 49.9) | −4.0 (−4.2, −3.8)* | 2467.1 (2060.7, 2905.8) | 732.6 (601.9, 876.7) | −4.1 (−4.4, −3.8)* |
| | Female | 70.4 (49.1, 94.7) | 22.5 (14.6, 31.7) | −3.9 (−4.1, −3.6)* | 1116.1 (855.4, 1393.3) | 331.9 (249.8, 430.0) | −4.1 (−4.3, −3.9)* |

*AAPC is significantly different from zero at the alpha=0.05 level.
AAPC, average annual percentage change; LDL, low-density lipoprotein; UI, uncertainty interval.

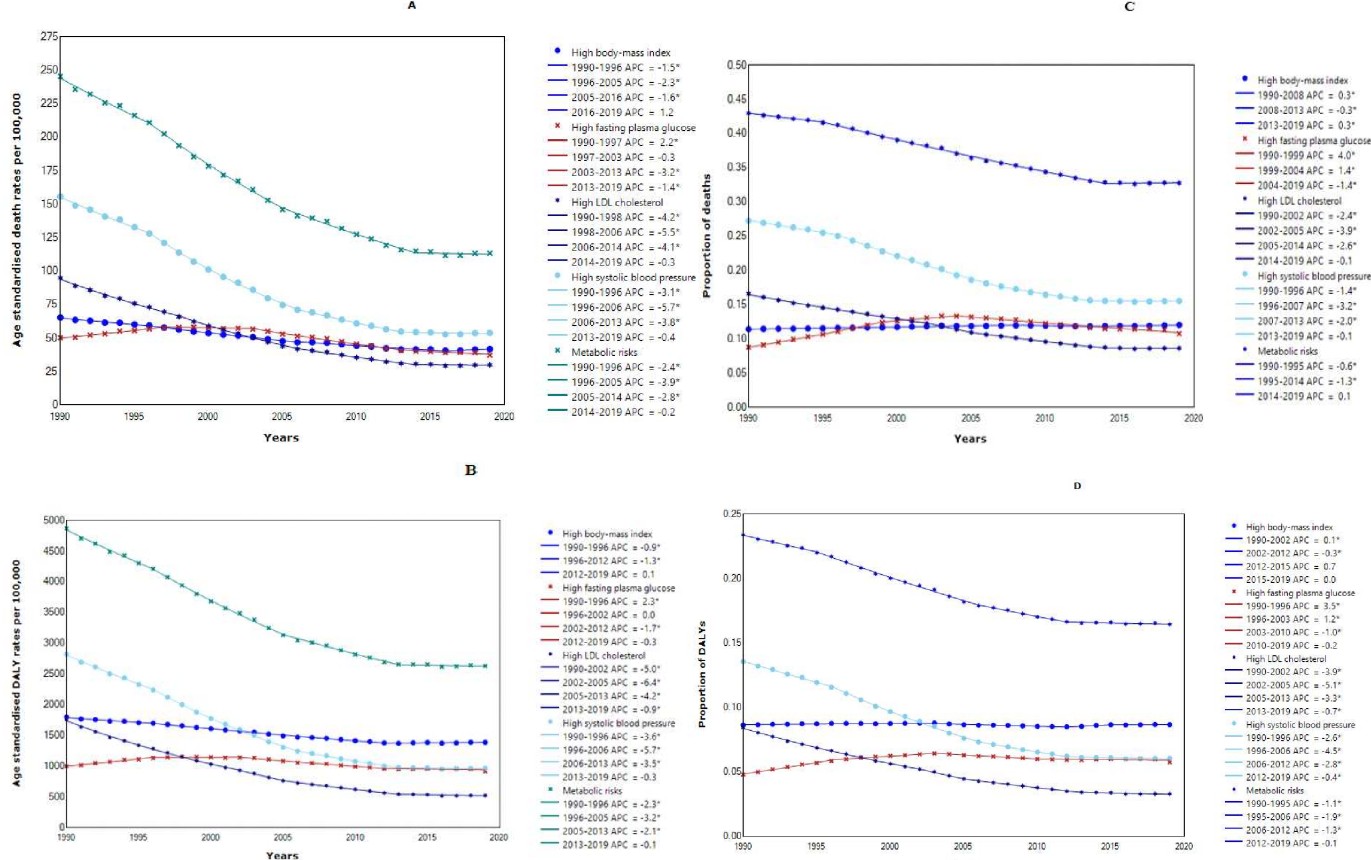

**Figure 1** Trends of NCDs deaths and DALYs attributable to combined and individual metabolic risks: age-standardised death rates (A); proportion of deaths (B); age-standardised DALY rates (C) and proportions of DALYs (D) between 1990 and 2019 in Australia from join point regression analysis. (Note: * indicates APC is significantly different from zero at the α=0.05 level). APC, annual percentage change; DALYs, disability-adjusted life-years; NCD, non-communicable disease.

Specifically, age-standardised death rates of CKD related to BMI and FPG significantly increased by 1.1% and 2.0% per year, and death rates of neurological disorders linked to BMI and FPG increased by 1.0% and 0.9% per year. In contrast, age-standardised CVD death rates attributable to high BMI, high FPG, high SBP and high LDL cholesterol decreased by 2.8%, 1.5%, 3.9% and 3.9% per year, respectively, between 1990 and 2019 (online supplemental table 5).

The metabolic risk-related age-standardised DALY rates for CVD and CRDs significantly decreased by 3.6% and 0.6% per year between 1990 and 2019, whereas metabolic risk-related DALY rates of neoplasms, neurological disorders, diabetes mellitus and CKD increased by 0.4%, 1.0%, 0.6% and 0.3% per year, respectively. However, the decline of metabolic risk-related CVD DALY rates between 2010 and 2019 (1.7% per year) were slow compared with over the last 30 years (3.6% per year) (online supplemental table 6).

Similarly, age-standardised DALY rates of CVD attributable to High BMI, high FPG, high SPB and high LDL cholesterol declined by 2.7%, 1.5%, 3.9% and 4.1% per year between 1990 and 2019. But between 2010 and 2019, the rate of decline has slowed with an average annual rate of change of 1.2%, 1.9%, 1.6% and 2.0% for high BMI,

high FPG, high SPB and high LDL cholesterol, respectively (online supplemental table 6).

## DISCUSSION

We assessed the trends and impact of metabolic risk factors on the burden of NCDs in the Australian population across age and sex between 1990 and 2019. Although the relative burden of metabolic risk-related NCDs in Australia has declined between 1990 and 2019, metabolic risks continue to account for a significant proportion, and absolute number, of the NCD burden. Approximately, one-third (34%) of NCD deaths and one-fifth (20%) of NCD DALYs were attributed to metabolic risk factors in 2019. Moreover, most CVD deaths and deaths from type 2 diabetes and CKDs were attributed to metabolic risks.

The ASR of deaths and DALYs attributed to metabolic risks decreased between 1990 and 2019, although the absolute number of metabolic risk-related deaths and DALYs have increased. The reductions in age standardised NCD burdens attributed to metabolic risks could be because of enhancement in prevention programmes, and the availability of effective treatments.[27–29] In addition, the increase in NCD burden due to increased prevalence of BMI and FPG could offset by reduced risks from SBP

**Table 2** Burden of specific NCDs (deaths and DALYs) attributable to metabolic risks in Australia, 2019

| Specific causes of NCDs | Deaths | | DALYs | |
|---|---|---|---|---|
| | Proportion, % (95% UI) | ASRs per 100000 (95% UI) | Proportion, % (95% UI) | ASR per 100000 (95% UI) |
| Combined metabolic risks | | | | |
| Neoplasms | 10.3 (6.0, 15.61) | 13.0 (7.5, 19.9) | 9.5 (5.7, 14.1) | 274.2 (164.4, 407.6) |
| Cardiovascular diseases | 68.1 (61.6, 73.8) | 73.6 (62.2, 82.8) | 68.3 (63.0, 72.9) | 1279.8 (1145.9, 1410.1) |
| Chronic respiratory diseases | 1.6 (1.0, 2.3) | 0.4 (0.2, 0.6) | 9.3 (6.1, 13.1) | 88.5 (54.1, 138.7) |
| Neurological disorders | 16.6 (5.9, 29.2) | 5.8 (1.0, 16.6) | 7.02 (2.5, 14.8) | 85.1 (26.4, 202.8) |
| Diabetes mellitus | 100.0 (100.0, 100.0) | 9.2 (8.2, 10.0) | 100.0 (100.0, 100.0) | 480.8 (365.6, 613.0) |
| Chronic kidney disease | 100.0 (100.0, 100.0) | 10.8 (9.0, 12.1) | 100.0 (100.0, 100.0) | 208.5 (184.8, 233.2) |
| High body mass index | | | | |
| Neoplasms | 6.7 (4.1, 9.5) | 8.4 (5.2, 12.0) | 6.6 (4.1, 9.2) | 190.3 (118.8, 265.9) |
| Cardiovascular diseases | 18.6 (11.8, 26.1) | 20.1 (12.7, 28.4) | 25.5 (17.6, 33.7) | 477.8 (327.8, 641.5) |
| Chronic respiratory diseases | 1.6 (1.0, 2.3) | 0.4 (0.2, 0.6) | 9.3 (6.12, 13.1) | 88.5 (54.1, 138.7) |
| Neurological disorders | 11.8 (3.8, 23.2) | 4.1 (0.7, 12.2) | 5.1 (1.5, 11.4) | 61.6 (17.4, 152.5) |
| Diabetes mellitus | 47.0 (31.9, 62.2) | 4.3 (2.9, 5.9) | 58.1 (44.9, 68.6) | 279.5 (192.1, 383.4) |
| Chronic kidney disease | 35.4 (16.5, 55.5) | 3.8 (1.8, 6.0) | 39.7 (25.4, 54.5) | 82.7 (52.6, 116.2) |
| High fast plasma glucose | | | | |
| Neoplasms | 3.8 (1.0, 7.9) | 4.8 (1.3, 9.9) | 31 (0.8, 6.4) | 89.1 (23.9, 184.0) |
| Cardiovascular diseases | 18.6 (11.0, 31.0) | 20.1 (11.8, 33.4) | 15.5 (10.1, 23.8) | 289.8 (186.2, 455.0) |
| Neurological disorders | 5.9 (1.1, 14.3) | 2.1 (0.2, 7.8) | 2.4 (0.4, 6.7) | 29.1 (5.1, 95.7) |
| Diabetes mellitus | 100.0 (100.0, 100.0) | 9.2 (8.2, 10.0) | 100.0 (100.0,100.0) | 480.8 (365.6, 613.0) |
| Chronic kidney disease | 9.5 (6.6, 13.6) | 1.0 (0.7, 1.5) | 13.9 (11.1, 17.0) | 29.1 (22.1, 37.3) |
| High systolic blood pressure | | | | |
| Cardiovascular diseases | 43.7 (36.2, 51.6) | 47.3 (37.5, 57.3) | 45.7 (39.6, 51.3) | 855.6 (720.6, 989.9) |
| Chronic kidney disease | 60.9 (52.7, 68.2) | 6.5 (5.3, 7.7) | 53.2 (46.0, 59.7) | 110.9 (92.3, 131.4) |
| High LDL cholesterol | | | | |
| Cardiovascular diseases | 27.71 (20.0, 36.3) | 30.0 (21.1, 40.1) | 28.0 (23.0, 33.9) | 525.2 (420.9, 639.0) |

ASR, age-standardised rate; DALYs, disability-adjusted life-years; LDL, low-density lipoprotein; NCDs, non-communicable diseases; UI, uncertainty interval.

and LDL. The rise in the absolute number of metabolic risk-related NCD burdens could be due to the growing population size and ageing population in Australia and advances in diagnostic methods that help diagnose the disease at its early stages.[30] The incidence of many NCDs increases with age and many older individuals now live with chronic health conditions.[6] In addition to the metabolic risk factors, other non-metabolic factors[16] might contribute to the NCDs burden in Australia. For instance, dietary risks and tobacco significantly contributed for the burden of NCD in 2018 and are risk factors themselves for the metabolic risks we examined in this study.[5]

We show that NCD deaths and DALYs related to metabolic risks differed substantially by sex. The burden of NCDs attributed to metabolic risks was higher for males than for females and decreased overtime for both sexes at different rate. This might be due to the difference in risk factor distributions among genders (eg, proportion of overweight is higher in males (75%) than females (60%), and males had higher smoking rates (16.5%) than females (11%).[6]

High BMI, SBP and FPG were responsible for high proportions of metabolic risk attributable NCD burden in 2019, each accountable for 10%–16% of age standardised NCD deaths, and 5%–8% of NCD DALYs. This is consistent with the results of the studies that show a high burden of NCD deaths was associated with high BMI and high SBP in Australian population in 2018.[5 16] Metabolic risk factors are one of the health risk factors targeted for NCD prevention in the current Australian NCD National Framework.[29] Metabolic risk factors along with other behavioural risk factors are important in Australia: nearly 90% of individuals over 45 years of age had at least one risk factor, and two-thirds of them had over three risk factors in 2012.[31] The prevalence of high BMI, SBP and FPG was also high in the Australian population in 2017/2018.[6]

Our analysis of 2019 GBD data showed that high SBP, BMI, high FPG and high LDL cholesterol were ranked first, third (behind smoking), fourth and fifth among the top five leading risk factors of NCD deaths in Australia, respectively.[32] BMI (first), SBP (third, behind smoking) and FPG (fourth) were the leading risk factors for NCD DALYs in the country.[32] This metabolic risk-related NCD burden is evident worldwide,[33 34] with evidence showing that a considerable proportion of NCD deaths is attributed to high blood pressure (13%), high blood sugar (6%) and overweight (5%).[35]

Our study showed that the proportion of the age standardised NCD death rates related to BMI and FPG slightly increased, while the proportion of NCDs attributable to SBP and LDL decreased, between 1990 and 2019. In line with these findings, BMI and FPG were the significant contributors to NCD deaths in Australia in 2018.[5 16] The recent Australian Health Survey also revealed that the prevalence of high BMI has increased from 63% in 2014–2015 to 67% in 2017–2018 in the adult population.[6] The other study findings also show that the proportions of individuals with obesity, high cholesterol and high blood

glucose increased between the year 2004 and 2013.[36] Suboptimal dietary intake and low physical activity levels are intermediary risk factors between the metabolic risks and NCDs.[35] Reducing the burden of NCDs will therefore require, in addition to better treatment and management of metabolic risks, population-based prevention actions focused on improving diets and increasing physical activity levels.[34]

Our study demonstrated that, between 1990 and 2019, age-standardised metabolic risk factors have increased. Particularly, exposure to high BMI, FPG and LDL cholesterol levels have increased over this period. Throughout this period of increasing metabolic risk exposure, however, age-standardised CVD deaths in Australia declined, which is in line with global trends towards decreasing age-standardised CVD deaths, as reported in.[4 18] The justification for the apparent paradox may be the effect of access to care and great improvements in treatment, sociocultural factors, cohort effects, and other behavioural and environmental factors not recorded in the GBD study.[37]

The findings revealed that in 2019, CVDs, neoplasm and CKD are the leading causes of NCDs deaths attributable to metabolic risk, and CVD, diabetes mellitus and neoplasms are the leading causes of NCD DALYs attributable to metabolic risks. Of metabolic risk factors, BMI was accountable for a substantial proportion of deaths and DALYs due to diabetes, CKD and CVD. This is in line with other study findings that metabolic risks have been a significant contributor for high burden of NCD, especially for CVD and diabetes in Australia[5] and globally.[33] Notably, overweight and obesity have been the main driver of mortality due to CVD, diabetes and chronic disease in Australia and the USA.[38] Worldwide, high BMI has an increasing trend and was responsible for a high burden of CVD and diabetes mellitus.[39 40]

The result of this study shows that age-standardised death rates of CVD related to combined metabolic risks have significantly decreased while age-standardised death rates of neurological disease and CKD related to combined metabolic risks have increased. This is in line with the evidence that shows deaths of CVD have substantially declined in Australia, while the burden of neurological disorders has increased.[5] In 2018, neoplasm (34%) and CVDs (21%) were the leading causes of death in Australia and high BMI and FPG were the main contributors to this burden.[5] In addition, CVD deaths and DALYs attributed to metabolic risks decreased significantly (3.5% and 3.6% per year, respectively) between 1990 and 2019, but the rate of decline slowed in the last decade (2010–2019) (1.9% and 1.7% per year). In Australia, there has been a recent increase in the prevalence of metabolic risk factors such as obesity and diabetes,[6] which may slow the decline in CVD deaths and DALYs. In addition, the ageing of the Australian population may also partly explain the slower decline in CVD deaths and DALYs over the past decade.[30]

The findings of this study provide the estimates of the level and trends of the metabolic risk-related NCD burden at the national level covering three decades. The

results underscore the importance of metabolic risks for the prevention of NCD burden. The prevention of NCDs requires a multifaceted strategy that targets the 'causes of the causes' of NCDs,[28] and prioritising prevention strategies for metabolic risk factors to improve population health. Reducing overweight in Australia needs to be comprehensive and wide-ranging, involving multisectoral collaboration and regulatory reforms.[41] The most recent National Obesity Strategy, launched in Australia in 2022, focuses on the addressing the causes of obesity to reduce the population prevalence of obesity. Australian governments (States, Territories and Commonwealth) should urgently implement the strategy[42] to address the rising absolute burden of NCDs. Further study should assess the spatiotemporal pattern of NCD burdens attributed to metabolic risks incorporating the sociocultural contexts to inform practices and strategies at the lower local levels.

This study has strengths and limitations. One of the strengths of this study is that we used a longitudinal statistical method (joinpoint regression analysis) to model the long-term trend of NCD burden attributed to metabolic risks. This modelling method helps to segment the long period (1990–2019) into parts and identify the changing pattern of NCD burden for each period interval. Another strength of this study is the use of data from the GBD dataset, which is one of the most comprehensive and reliable data sources.

The limitations of this study include the likelihood that the metabolic risk-related disease burden may vary between rural and urban areas, different regions or states and sociocultural conditions, which could not be characterised in this study. In GBD, the use of similar relative risks in certain age and gender groups in all countries may overestimate or underestimate the exact picture of disease burden, because metabolic risk factors may not influence disease outcomes in the same way in different population groups.[4] Additionally, because the GBD study used secondary data, there might be a lack of metabolic risk data to estimate exposure levels, although this was addressed by the modelling methods using spatiotemporal Gaussian process regression and DisMod meta-regression V.2.1(4).

## CONCLUSION

Despite a relative reduction in the metabolic risk-related burden of NCDs in Australia between 1990 and 2019, metabolic risks continue to contribute significantly to the burden of NCDs, primarily CVD, neoplasms, diabetes mellitus and CKD in the country. The absolute burden of NCDs also continues to increase. This study adds to the growing body of evidence that metabolic risk factors are important contributors to the burden of NCDs in Australia and underscores the need for effective prevention and management strategies to address these risk factors. To reduce the substantial burden of NCDs, multifaceted strategy and policies with multisectoral collaboration should focus on metabolic risks and other upstream NCD risk factors.

**Acknowledgements** The authors would like to thank the Institute for Health Metrics and Evaluation for providing the data.

**Contributors** Conceptualised and designed the study: KTK and MN; analysed the data: KTK;guarantor: KTK; interpreted the results: KTK, MN, AP, KB and FT; drafted the manuscript: KTK and MN. Revised the manuscript critically for important intellectual content: KTK, MN, AP, KB and FT. All authors have read and approved the manuscript.

**Funding** This research received no specific grant from any funding agency or the commercial or not-for-profit sectors. KTK is supported by Dean's Postdoctoral Fellowship at Deakin University (N/A). KB is supported by a Heart Foundation Future Leader Fellowship (102047). AP is supported by a National Health and Medical Research Council (NHMRC) Investigator Grant (N/A). MN is supported by an NHMRC Ideas Grant (GNT2002334).

**Disclaimer** The contents of this publication are solely the responsibility of the authors and do not reflect the views of the NHMRC.

**Competing interests** None declared.

**Patient and public involvement** Patients and/or the public were not involved in the design, or conduct, or reporting, or dissemination plans of this research.

**Patient consent for publication** Not applicable.

**Provenance and peer review** Not commissioned; externally peer reviewed.

**Data availability statement** All data relevant to the study are included in the article or uploaded as online supplemental information. 'Not applicable'.

**ORCID iD**
Kelemu Tilahun Kibret http://orcid.org/0000-0002-4357-4122

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
