## [Reviewer comments · BMJ Open]

ARTICLE DETAILS

TITLE (PROVISIONAL)	The burdens of non-communicable disease attributable to metabolic risk factors in Australia, 1990–2019: Joinpoint Regression Analysis of the Global Burden of Disease Study
AUTHORS	Kibret, Kelemu; Backholer, Kathryn; Peeters, A; Tesfay, Fisaha; Nichols, Melanie

VERSION 1 – REVIEW

REVIEWER	Olatona, Foluke University of Lagos College of Medicine, Community Health and Primary Care
REVIEW RETURNED	01-Mar-2023

GENERAL COMMENTS	Review Comments on “Trends and Impact of Metabolic Risk Factors on NCD burdens in Australia, 1990 – 2019: Analysis of the Global Burden of Disease Study” Areas of Improvement/Weaknesses • Abstract The abstract could have included a brief background or introduction in addition to the stated objective State the period of the study clearly not just three decades• Methods: Add study design, sample size, study location, and target population. The acronym “NCD” was omitted with the first use of the phrase “Non-Communicable Diseases” within the Abstract• Methods : Add key information on study design and study size• Results • The “Age-standardised death rates and disability-adjusted life years (DALYs) rates of non-communicable diseases (NCDs) attributable to metabolic risks and their temporal trends between 1990 to 2019 by sex” dataset of Table 2 could be better spaced to make it more reader-friendly.
---

REVIEWER	Cissé, Kadari Institut de Recherche en Sciences de la Sante, Biomedical et santé publique
REVIEW RETURNED	23-Apr-2023

GENERAL COMMENTS	Thank you for sharing this article with me. This is an interesting paper on Trends and impacts of metabolic risk factors on noncommunicable disease burdens in Australia on a long time period (1990-2019). The manuscript is well written. The objective, methods, results and discussion are well described and presented. Major comments
--

The added value of this study in line with GBD 2019 study is not very clear for me. There is any context specific relation between metabolic risk factors and NCD which need to be highlighted?? Which new knowledge about the link between metabolic risk factors and NCD would you like the share??

Furthermore, in many results, authors have compared the trend of two periods: 1990-2010 and 2010-2019. However, it is not clear in the manuscript why authors have chosen these two periods. There were any NCD specific policies or actions in 2010 which may explain this choice. Moreover, authors have shown that there is low decreasing in some indicators between 2010 and 2029. In the discussion, they have not provided an explanation for slowing the decrease in this period.

There are too many results in the manuscript which tend to confuse reader.

Minor comments:

Abstract: author told that “their role in NCD death and disability remain a challenge as prevalence of these risk factors have increased”. This conclusion is not supported by results in the abstract. I suggest to add results which show the increasing prevalence?

Page 8: line 10-12: authors stated that “The age-standardised SEV of combined metabolic risks among the Australian population increased from 24.0 (95% UI: 19.1 - 29.7) in 1990 to 33.2 (95% UI: 27.4 -39.3) in 2019”. However, when we look at the UI it seems to overlap which mean no statistical difference between prevalence. might author clarify this?

Table 1: It is not clear for me why author decided to stratified the burden of NCDs attributable to metabolic risk factors but not the risk exposure for metabolic risk factors (supplement Table 1). There is any raison?

Page 14: “The age-standardised proportion of diabetes mellitus and chronic kidney disease deaths was 100% attributed to metabolic risks in Australia.” It is strange that 100% of CKD death was attributable to metabolic risks factors since there many other risk factors of CKD.

Page 14 line 22-25: authors stated that “About 47.0% deaths and 58.1% DALYs of diabetes mellitus, and about 35.4% deaths and 39.7% DALYs of CKD were attributed to high BMI”. However, in line 28-29 they also stated that “The results show that 100% diabetes mellitus deaths and DALYs was attributable to high FPG.” These two sentences seem to be conflicting. I thing that it will be clearer if author can provide death of diabetes mellitus and CKD attributable to combined metabolic risk factors (like BMI and FPG).

Page 20 line8-12: “However, the decline of metabolic risk-related CVD DALY rates in the past decade (2010-2019) (1.7% per year) were slow compared to over the last 30 years (3.6% per year) (Table 5). is There any raison for this difference in the trend?

Page 26 line 7-12: “This modelling method helps to segment the long period into parts and identify the changing pattern of NCD burden for each period interval”. It is not clear how and why author have chosen periods.

	Thank you.
--	------------

VERSION 1 – AUTHOR RESPONSE

Reviewer 1 comments			
Abstract			
 • The abstract could have included a brief background or introduction in addition to the stated objective 	We have revised and formatted the abstract according to the instructions for authors as above, and now include these sections.	2	1-4
 • State the period of the study clearly not just three decades 	The time period is now clearly stated as 1990 - 2019 or between 1990 and 2019.	2	4,11,15
 • Methods: Add study design, sample size, study location, and target population. 	It has been revised accordingly	2	5-11
 • The acronym “NCD” was omitted with the first use of the phrase “Non-Communicable Diseases” within the Abstract 	NCD has now been spelt in full (Non-communicable diseases)	2	13
Methods			
 • Add key information on study design and study size 	We have added information about study design This study is a population based observational study with an in-depth analysis of the Global Burden of Diseases study 2019 data of Australia using a Joinpoint regression analysis.	5	11-13
Results			
 • The “Age-standardised death rates and disability-adjusted life years (DALYs) rates of non-communicable diseases (NCDs) attributable to metabolic risks and their temporal trends between 1990 to 2019 by sex” dataset of Table 2 could be better spaced to make it more reader-friendly. 	It has been revised accordingly and please note that it is Table 1 in the revised version of the manuscript	10 and 11	
Reviewer 2 comments			
1. Thank you for sharing this article with me. This is an interesting paper on Trends and impacts of metabolic risk factors on noncommunicable disease	Thank you.		

burdens in Australia on a long time period (1990-2019). The manuscript is well written. The objective, methods, results and discussion are well described and presented.	
Major comment	
2. The added value of this study in line with GBD 2019 study is not very clear for me. There is any context specific relation between metabolic risk factors and NCD which need to be highlighted?? Which new knowledge about the link between metabolic risk factors and NCD would you like the share??	Thank you for this comment. We have made a number of revisions through the manuscript to better highlight the novelty and significance of the study. These are as follows: 4 18-23 Australia is a high-income country that has seen an overall decline in cardiovascular disease (CVD) mortality rates for some years now (14), but continues to experience a high prevalence of metabolic risk factors (5). Australia's situation is not unique - many high-income countries also face similar challenges (15). For example, the United States, and many European countries also have high levels of obesity (16) and type 2 diabetes (17). 5 2-7 This study provides context-specific insights into the burden of NCDs attributable to key metabolic risk factors in Australia. The study demonstrates the changing trends in this burden of NCDs attributable to metabolic risk factors over time. Our findings can inform the development of targeted prevention and management strategies for specific population groups, defined by defined by high BMI and high blood pressure. 19 19-21 This study adds to the growing body of evidence that metabolic risk factors are important contributors to the burden of NCDs in Australia and underscores the need for effective prevention and management strategies to address these risk factors
3. Furthermore, in many results, authors have compared the trend of two	Thank you for the comment. Our choice of 2010 -2019 period was

periods: 1990-2010 and 2010-2019. However, it is not clear in the manuscript why authors have chosen these two periods. There were any NCD specific policies or actions in 2010 which may explain this choice.	because we wanted to see the trends of a recent decade, which is of policy relevance, as key policies and strategies related to NCDs have been published in the last decade. We have updated the manuscript to further clarify the reasoning behind this decision and focus on more recent years, as follows: The average annual percentage change (AAPC) with its 95% CI for the entire period (1990 -2019) and for the last decade (2010-2019) was also calculated, with respect to the underlying model fitted by the Joinpoint analysis. We examined the trends of the most recent decade for which data were available (2010-2019) to more closely examine the recent trends, which are of policy relevance, and because key policies and strategies related to NCDs have been published in the last decade, such as a Global Action Plan for the Prevention and Control of NCDs (8) and National Primary Health Care Strategy of Australia (25).	8 1-8
4. Moreover, authors have shown that there is low decreasing in some indicators between 2010 and 2019. In the discussion, they have not provided an explanation for slowing the decrease in this period.	We have added the following discussion of the possible explanation: In addition, CVD deaths and DALYs attributed to metabolic risks decreased significantly (3.5% and 3.6% per year, respectively) between 1990 and 2019, but the rate of decline slowed in the last decade (2010–2019) (1.9% and 1.7% per year). In Australia, there has been a recent increase in the prevalence of metabolic risk factors such as obesity and diabetes (6), which may slow the decline in CVD deaths and DALYs. In addition, the ageing of the Australian population may also partly explain the slower decline in CVD deaths and DALYs over the past decade (30).	18 11-17
5. There are too many results in the manuscript which tend to confuse reader.	We have revised the results section to make it clearer and moved some results to the supplementary information. Table 1 ,Table 3, Table 4, Figure 2 and Figure 3 have been	8-15

	moved into supplementary information.		
Minor comments:			
6. Abstract: author told that “their role in NCD death and disability remain a challenge as prevalence of these risk factors have increased”. This conclusion is not supported by results in the abstract. I suggest adding results which show the increasing prevalence?	The abstract text has been revised to include the results on which this conclusion was based. The following was added to the results section of the abstract: Results indicate a 1.1% yearly increase in exposure to combined metabolic risk factors from 1990 to 2019.	2	10-11
7. Table 1: It is not clear for me why author decided to stratified the burden of NCDs attributable to metabolic risk factors but not the risk exposure for metabolic risk factors (supplement Table 1). There is any raison?	Supplementary Table 1 presents the summary exposure value (SEV), which is a statistical measure used to summarize the overall level of exposure to metabolic risk factors in a population. While table 1 presents the age standardised death and DALY rates of NCDs attributable to metabolic risk factors.		
8. Page 14: “The age-standardised proportion of diabetes mellitus and chronic kidney disease deaths was 100% attributed to metabolic risks in Australia.” It is strange that 100% of CKD death was attributable to metabolic risks factors since there many other risk factors of CKD.	Thanks for this comment, and we agree it can be confusing. Further explanation of why the GBD data are structured this way has been incorporated into the manuscript. Here the age standardised proportion is population attributable fraction (PAF). For some diseases or outcomes in the Global Burden of Disease (GBD) study, certain risk factors may have a PAF of 100%. The PAF can range from 0% to 100%, but a PAF of 100% does not necessarily mean that the risk factor or exposure is the sole cause of the health outcome or disease (4). With the case of diabetes and CKD, metabolic risks (high fasting plasma glucose and kidney dysfunction), explain 100% of the deaths related to diabetes and CKD, respectively (4). However, this does not necessarily mean that the FPG and kidney dysfunction are the sole cause of diabetes and CKD death, respectively.	7	9-18
9. Page 14 line 22-25: authors stated that “About 47.0% deaths and 58.1% DALYs of diabetes mellitus, and about	It has been described as outline above (comment # 8)	7	9-18

35.4% deaths and 39.7% DALYs of CKD were attributed to high BMI". However, in line 28-29 they also stated that "The results show that 100% diabetes mellitus deaths and DALYs was attributable to high FPG." These two sentences seem to be conflicting. I think that it will be clearer if author can provide death of diabetes mellitus and CKD attributable to combined metabolic risk factors (like BMI and FPG).	Here the age standardised proportion is population attributable fraction (PAF). For some diseases or outcomes in the Global Burden of Disease (GBD) study, certain risk factors may have a PAF of 100%. The PAF can range from 0% to 100%, but a PAF of 100% does not necessarily mean that the risk factor or exposure is the sole cause of the health outcome or disease (4). With the case of diabetes and CKD, metabolic risks (high fasting plasma glucose and kidney dysfunction), explain 100% of the deaths related to diabetes and CKD, respectively (4). However, this does not necessarily mean that the FPG and kidney dysfunction are the sole cause of diabetes and CKD death, respectively. There are no available estimates for the combined BMI and FPG	
10. Page 20 line8-12: "However, the decline of metabolic risk-related CVD DALY rates in the past decade (2010-2019) (1.7% per year) were slow compared to over the last 30 years (3.6% per year) (Table 5). is There any reason for this difference in the trend?"	We have added the explanation in the discussion In addition, CVD deaths and DALYs attributed to metabolic risks decreased significantly (3.5% and 3.6% per year, respectively) between 1990 and 2019, but the rate of decline slowed in the last decade (2010–2019) (1.9% and 1.7% per year). In Australia, there has been a recent increase in the prevalence of metabolic risk factors such as obesity and diabetes (6), which may slow the decline in CVD deaths and DALYs. In addition, the ageing of the Australian population may also partly explain the slower decline in CVD deaths and DALYs over the past decade (30)	18 11-17
11. Page 26 line 7-12: "This modelling method helps to segment the long period into parts and identify the changing pattern of NCD burden for each period interval". It is not clear how and why author have chosen periods.	Thank you for this comment. We have addressed this question and expanded our justification of the time periods It has been now described in the method section at page 7 as follows: Using these extracted data from GBD results tool, the temporal trends in NCD burdens (deaths and DALYs) attributable to each metabolic risk	7 30-31

	factor over time (1990 to 2019) were assessed using the Joinpoint regression model (separately by sex) (19). This regression model segments the entire period and identifies the time points (years) at which there has been a significant change in trends (Joinpoints) (20). Then, the annual percentage change (APC) with its 95% confidence interval (CI) of rates between these inflection points is estimated. The average annual percentage change (AAPC) with its 95% CI for the entire period (1990 - 2019) and for the last decade (2010-2019) was also calculated, with respect to the underlying model fitted by the Joinpoint analysis. We examined the trends of the most recent decade for which data were available (2010-2019) to more closely examine the recent trends, which are of policy relevance, and because key policies and strategies related to NCDs have been published in the last decade, such as a Global Action Plan for the Prevention and Control of NCDs (8) and National Primary Health Care Strategy of Australia (25)	8 1-11
--	--	---------------

VERSION 2 – REVIEW

REVIEWER	Cissé, Kadari Institut de Recherche en Sciences de la Sante, Biomedical et santé publique
REVIEW RETURNED	16-Jun-2023
GENERAL COMMENTS	Authors have satisfactorily addressed all my comments. They have significantly improved the manuscript. I do not have further comments. Thank you.

VERSION 2 – AUTHOR RESPONSE